# From Static Constraints to Dynamic Adaptation: Sample-Level Constraint Release for Offline-to-Online RL

## Abstract

Offline-to-online reinforcement learning (off2on RL) integrates the sample efficiency of offline pretraining with the adaptability of online fine-tuning. However, it suffers from a constraint-release dilemma: conservative objectives inherited from offline training ensure stability yet hinder adaptation, while uniformly discarding them induces instability. Existing approaches impose global constraints across all samples, thereby overlooking the distributional heterogeneity wherein offline and online data gradually overlap. We propose Dynamic Alignment for RElease (DARE), a distribution-aware framework that enforces the constraints at the sample level in a behavior-consistent manner. To this end, DARE employs a diffusion-based behavior model with energy guidance to generate reference actions, assigns alignment scores to individual samples, leverages Gaussian fitting to distinguish offline-like from online-like data, and exchange behavior-inconsistent samples between offline and online batches to ensure behavior-consistent constraint enforcement. We theoretically prove that DARE reduces offline–online distributional discrepancy while ensuring that value estimation errors remain bounded. Our empirical results on the D4RL benchmark demonstrate that integrating DARE into representative off2on methods (Cal-QL and IQL) consistently improves policy performance and achieves stable, robust, and adaptive fine-tuning. (Anonymized code archive is included in the supplementary material)

## 1 Introduction

Reinforcement learning (RL) has achieved remarkable progress in diverse sequential decision-making domains, such as recommendation systems (Wang et al., 2018; Chen et al., 2024), robotics (Rafailov et al., 2023; Zhao et al., 2023), and large language models (Du et al., 2023; Zhai et al., 2024). These successes have been driven mainly by two learning paradigms: online RL, which learns through direct interactions with the environment, and offline RL, which relies solely on pre-collected datasets. Online RL can reach high performance but requires massive environment interactions (Mnih et al., 2015; Song et al., 2023; Alonso et al., 2024). Offline RL, in contrast, avoids these risks by training on static datasets, offering higher sample efficiency but limited adaptability to unseen situations (Fujimoto et al., 2019; Kumar et al., 2020; Uehara et al., 2024). To combine their complementary strengths, recent work has explored the offline-to-online (off2on) RL paradigm (Xie et al., 2021; Zhang & Zanette, 2024), where an agent is first pretrained on offline data and then fine-tuned through a small number of online interactions (Lee et al., 2022; Kostrikov et al., 2021; Nakamoto et al., 2023). However, when the agent explores beyond the offline dataset, it encounters out-of-distribution (OOD) issue, which often causes value overestimation and policy drift. This issue, coupled with the conservative loss introduced during offline training, hinders exploration and adaptation while removing it abruptly can destabilize learning (Ball et al., 2023).

To address this constraint-release dilemma, existing off2on approaches are mostly built upon specific offline algorithms and fall into two main categories, Policy-constraint and Value-regularization. Policy-constraint methods limit the policy to stay close to the offline behavior (Wu et al., 2022; Zhang et al., 2023), which helps maintain stability but often suppresses exploration and slows improvement. Value-regularization methods (Nakamoto et al., 2023), such as CQL (Kumar et al., 2020), encourage conservative value estimates to avoid overestimation, but suffer from unstable Q-

value fluctuations and high computational cost (Lee et al., 2022; Zhang et al., 2024b). More recent studies attempt to ease the transition from offline pretraining to online fine-tuning by introducing auxiliary components (Wang et al., 2023), such as aligning the actor and critic to reduce policy-value inconsistency (Luo et al., 2024), and employing generative models to enrich training data (Liu et al., 2024). Despite these advances, most methods apply constraints uniformly across all samples, regarding the offline dataset merely as initialization and neglecting its distributional structure (Mao et al., 2024). In practice, however, offline and online data gradually overlap during fine-tuning, with some offline samples exhibiting exploratory behavior and some online samples remaining aligned with the offline policy. Failing to account for this heterogeneity results in constraints that are either overly conservative or released too aggressively.

To fill this gap, we propose Dynamic Alignment for RElease (DARE), an offline-to-online RL framework that enforces constraints in a sample-specific and behavior-consistent manner. Because offline and online data distributions overlap, dataset labels (offline vs. online) are insufficient to determine whether a sample exhibits offline-like or online-like behavior. DARE addresses this issue by evaluating each sample's behavioral alignment and apply constraints accordingly. Specifically, DARE leverages a diffusion-based generative model to generate reference actions. These reference actions allow us to assign alignment scores to individual samples, quantifying their behavioral consistency with the offline policy. By fitting the alignment distributions with Gaussian models, we derive a data-driven rule that distinguishes offline-like from online-like samples. These behaviorally inconsistent samples are then adaptively exchanged between offline and online batches, ensuring that constraints are preserved or relaxed in a behavior-consistent manner. In the end, we integrate DARE into two representative off2on algorithms, Cal-QL and IQL, demonstrating that it can be seamlessly applied by enforcing conservative objectives on offline-like samples while relaxing constraints for online-like ones.

We provide theoretical analysis of DARE, showing that our exchanges mechanism monotonically narrows the distributional discrepancy and that both the residual error and value estimation error remain bounded. Our ablation studies further dissect the contribution of each component, including the exchange budget and alignment mechanism. The empirical evaluations on the D4RL benchmark demonstrate that DARE enhances fine-tuning robustness and stability, achieving superior policy performance.

## 2 RELATED WORKS

**Offline-to-online Reinforcement Learning.** A critical challenge in off2on RL lies in accurate value estimation under out-of-distribution (OOD) actions and in mitigating the distributional shift that emerges during fine-tuning. To address the challenge, several methods aim to reduce Q-value bias. For example, SO2 (Zhang et al., 2024b) introduces perturbed updates to smooth value estimates, and SUF (Feng et al., 2024) adjusts the update-to-data (UTD) ratio to mitigate overfitting to static datasets. Another line of work focuses on distributional shift. Cal-QL (Nakamoto et al., 2023) and FamO2O (Wang et al., 2023) calibrate Q-values with offline data and progressively update them online, while Off2On (Lee et al., 2022) balances conservative pretraining with exploratory fine-tuning. Ball et al. (Ball et al., 2023) employ Layer Normalization to stabilize Q-value learning and prevent over-extrapolation during online fine-tuning. In parallel, PEX (Yu & Zhang, 2023) mitigates distributional shift by expanding the offline policy with additional exploratory components. More recent approaches such as OCR-CFT (Luo et al., 2024) and OPT (Shin et al., 2025) go beyond conventional fine-tuning. They reconstruct critics, align them with the policy, and then switch to purely online optimization, effectively increasing the update-to-data (UTD) ratio. While these methods advance stability and adaptability, they typically enforce constraints globally or rely on auxiliary heuristics, overlooking the heterogeneous overlap between offline and online samples.

**Diffusion Model in Reinforcement Learning.** Diffusion models have recently been adopted in RL as powerful generative tools. Diffuser (Janner et al., 2022) employs a diffusion model to generate entire action trajectories, guided by a separately trained return predictor to bias sampling toward high-return behaviors. SYNTHER (Lu et al., 2023b) extends this idea to both offline and online RL by using diffusion-based data upsampling to enrich training distributions. Other works exploit classifier- or function-guided diffusion for more targeted generation. For example, PolyGRAD (Rigter et al., 2023) embeds policy information into a classifier-guided diffusion model for on-policy world mod-

eling, while Chen et al. (Chen et al., 2022) leverage reward or Q-functions to guide diffusion sampling toward task-relevant behaviors. CEP (Lu et al., 2023a) approximates energy-based guidance through contrastive learning, focusing on offline data generation, and EDIS (Liu et al., 2024) extends energy-guided diffusion to the off2on setting by steering action sampling during fine-tuning. In contrast to these approaches, which use diffusion for trajectory or data generation, we explicitly model the offline behavior policy itself and leverage it to guide constraint release during fine-tuning.

# 3 PRELIMINARIES

## 3.1 OFFLINE-TO-ONLINE REINFORCEMENT LEARNING

RL is a framework in which an agent learns to maximize cumulative rewards by interacting with an environment (Mnih et al., 2013; Van Hasselt et al., 2016). The problem is modeled as a Markov Decision Process (MDP), defined by a tuple $(\mathcal{S}, \mathcal{A}, P, R, \gamma)$, where $\mathcal{S}$ is the state space, $\mathcal{A}$ is the action space, $P$ is the transition probability, $R$ is the reward function, and $\gamma \in [0, 1)$ is the discount factor. At each timestep $t$, the agent observes a state $s_t$, takes an action $a_t \in \mathcal{A}$, receives a reward $r_t = R(s_t, a_t)$, and then transitions to a new state $s_{t+1}$ according to $P(s_{t+1}|s_t, a_t)$.

In offline RL, the agent learns solely from a static dataset $\mathcal{D}_{off}$ collected by unknown policy $\nu$, without additional interaction with the environment (Fujimoto & Gu, 2021). The off2on RL extends this setting by further fine-tuning the policy with limited online interaction. During fine-tuning, the agent must balance knowledge from offline data with new online experiences $\mathcal{D}_{on}$, adapting the policy while avoiding overfitting to OOD actions. The replay buffer then becomes $\mathcal{D} = \mathcal{D}_{off} \cup \mathcal{D}_{on}$.

## 3.2 CALIBRATED Q-LEARNING

Calibrated Q-learning (Cal-QL) (Nakamoto et al., 2023) aims to learn a conservative and calibrated value function from an offline dataset. Cal-QL builds on CQL (Kumar et al., 2020) and constrains the learned Q-function to produce Q-values that are larger than the Q-value of a reference policy $\nu$:

$$
\begin{aligned}
\mathcal{L}_{\text{Cal-QL}}(Q) = &\frac{1}{2} \cdot \mathbb{E}_{(s,a,s') \sim \mathcal{D}} \left[ \left( Q(s,a) - \hat{\mathcal{B}}^\pi \hat{Q}_{\text{target}}(s,a) \right)^2 \right] \\
&+ \alpha \underbrace{\left( \mathbb{E}_{s \sim \mathcal{D}, a \sim \pi} \left[ \max(Q(s,a), V^\nu(s)) \right] - \mathbb{E}_{(s,a) \sim \mathcal{D}} \left[ Q(s,a) \right] \right)}_{\mathcal{R}}.
\end{aligned}
\tag{1}
$$

Here, $\alpha$ controls the strength of conservatism. We denote the second term in Eq. 1 as $\mathcal{R}$, the regularizer. During online fine-tuning, $\mathcal{R}$ naturally reduces to the standard CQL penalty.

## 3.3 IMPLICIT Q-LEARNING

Implicit Q-Learning (IQL) (Kostrikov et al., 2021) avoids Q-value overestimation on OOD actions by using expectile regression, $L_\tau^2(u) = |\tau - \mathbb{1}(u < 0)|u^2$. The value function is learned by minimizing the expectile regression loss:

$$
\mathcal{L}_{IQL}(V) = \mathbb{E}_{(s,a) \sim \mathcal{D}} \left[ L_\tau^2 \left( \hat{Q}_{\text{target}}(s,a) - V(s) \right) \right],
\tag{2}
$$

The Q-function is then updated via:

$$
\mathcal{L}_{IQL}(Q) = \mathbb{E}_{(s,a,s') \sim \mathcal{D}} \left[ \left( r(s,a) + \gamma V(s') - Q(s,a) \right)^2 \right].
\tag{3}
$$

IQL extracts the policy through advantage-weighted regression (Nair et al., 2020), where the learned Q-function is used to prioritize actions with higher advantages:

$$
\mathcal{L}_{IQL}(\pi) = \mathbb{E}_{(s,a) \sim \mathcal{D}} \left[ \exp \left( \beta \left( \hat{Q}_{\text{target}}(s,a) - V(s) \right) \right) \log \pi(a|s) \right].
\tag{4}
$$

# 4 MOTIVATION

We start by analyzing the distributional shift that occurs during the offline-to-online transition, as it represents a major challenge to achieve stable fine-tuning. Accordingly, we compute the mean squared error (MSE) between the actions predicted by the offline-pretrained policy and those

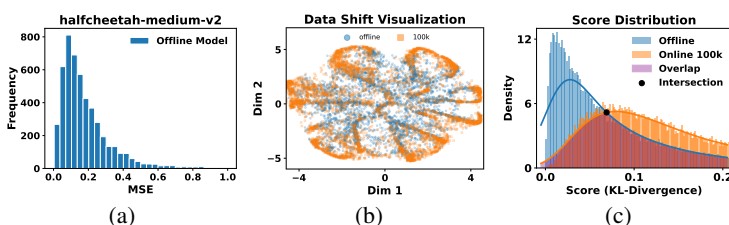

(a)  (b)  (c)

Figure 1: Characteristics of distributional shift in fine-tuning.

in the offline dataset. Fig. 1a shows the histogram of MSE values on *HalfCheetah-medium-v2*, indicating a significant mismatch between the policy outputs and dataset actions. As online training continues, such discrepancies can exacerbate into a considerable distributional shift. Fig. 1b illustrates this via a t-SNE projection: after 100k steps, online samples (orange) and offline data (blue) occupy distinct regions in feature space. While t-SNE is qualitative, such separation in projection suggests that offline and online distributions diverge substantially. These observations imply that applying a uniform loss to all samples is suboptimal, since some remain close to the offline distribution while others have shifted OOD. To further investigate this during fine-tuning, we approximate the offline action distribution with a behavior model and use it to score both offline and online samples via KL-based alignment. As shown in Fig. 1c, the distributions of these two scores exhibit a huge overlap. Such observations highlight the need for a reliable sample-level criterion to evaluate whether a data point is offline-like or online-like, so that training constraints can be applied in a behavior-consistent manner.

## 5    DARE: DYNAMIC ALIGNMENT FOR RELEASE

We propose **DARE**, a framework for offline-to-online reinforcement learning (RL) that enforces constraints adaptively, accounting for sample-specific characteristics and behavioral consistency with the offline dataset. Specifically, DARE generates reference actions via an energy-guided diffusion model that is guided by Q-function estimates. It then calculates alignment scores to quantify each sample's consistency with the offline data, and deploys Gaussian fitting to derive a data-driven threshold that separates offline-like from online-like samples. Through this process, DARE preserves conservative constraints on offline-like samples while adaptively relaxing them for exploratory ones.

### 5.1    REFERENCE ACTIONS THROUGH ENERGY-GUIDED DIFFUSION MODEL

Energy-guided diffusion has been widely deployed in generative modeling to improve data quality or sample efficiency (Lu et al., 2023a; Liu et al., 2024; Zhang et al., 2024a; Xu et al., 2024). Here, we adapt this formulation to the off2on RL setting in order to facilitate constraint release. Specifically, we construct *reference actions* that preserve the offline behavior while incorporating value information from the Q-function, enabling sample-level alignment estimation during fine-tuning.

Mathematically, we take the pretrained diffusion behavior model $\nu_0$ as the base distribution and use the Q-function as the energy term, which defines an induced policy $\pi_0$. The marginal distribution of a noise-perturbed action $a_t$ at time $t$ is

$$\pi_t(a_t \mid s) \propto \nu_t(a_t \mid s) \, e^{\mathcal{E}_t(s, a_t)}, \tag{5}$$

where $\mathcal{E}_t(s, a_t)$ is parameterized by a neural network $f_\phi(s, a_t, t)$ derived from $Q(s, a_0)$ and updated during fine-tuning. The corresponding score function is

$$\nabla_{a_t} \log \pi_t(a_t \mid s) = \nabla_{a_t} \log \nu_t(a_t \mid s) + \nabla_{a_t} \mathcal{E}_t(s, a_t). \tag{6}$$

This construction results in an energy-guided policy $\pi_0$, from which reference actions $\hat{a} \sim \pi_0(\cdot \mid s)$ are sampled.

## 5.2 ALIGNMENT SCORES VIA KULLBACK–LEIBLER DIVERGENCE

During the online phase, each training batch $b = b_{off} \cup b_{on}$ contains samples from both the offline dataset $\mathcal{D}_{\text{off}}$ and the online replay buffer $\mathcal{D}_{on}$. However, the dataset source alone does not indicate whether a sample exhibits offline-like or online-like behavior. An offline sample may reflect online exploratory behavior, while an online sample may still follow the offline distribution.

To capture this distinction, we assign each sample an *alignment score* that measures its consistency with the offline behavioral policy. For each state $s_i$ in $b$, we generate a reference action $\hat{a}_i \sim \pi_0(\cdot \mid s_i)$ using the energy-guided diffusion model in Eq. 5 and Eq. 6. The alignment score of the observed pair $(s_i, a_i)$ is then defined as the KL divergence between the actual action and the reference action:

$$\text{Align}(s_i, a_i) = D_{\text{KL}}\big(a_i \,\|\, \hat{a}_i\big). \tag{7}$$

A higher alignment score indicates that the sample deviates more from the offline behavioral policy. This score is used to distinguish offline-like from online-like samples during fine-tuning.

## 5.3 SAMPLE-LEVEL EXCHANGE BASED ON GAUSSIAN FITTING

To apply proper constraints for each sample, we first classify the sample as offline-like or online-like based on the alignment scores given in Eq. 7. The major challenge of determining a distinct separation point is that the score distributions of offline and online samples often overlap as shown in Fig. 1c. In addition, the data exhibit different OOD patterns across different batches, making it impractical to apply a single separation point for all batches. To tackle this issue, for each batch, we approximate the distributions of offline scores $\{d_i^{\text{off}}\}$ and online scores $\{d_i^{\text{on}}\}$ with Gaussian models parameterized by empirical means $(\mu_{off}, \mu_{on})$ and standard deviations $(\sigma_{off}, \sigma_{on})$. While Gaussian models do not capture all distributional details, they provide a light-weight yet stable approximation that avoids overfitting to empirical distributions and results in interpretable decision boundaries.

The separation point $\tau$ is determined by the intersection of two Gaussian probability density functions (PDF):

$$\phi(\tau; \mu_{off}, \sigma_{off}^2) = \phi(\tau; \mu_{on}, \sigma_{on}^2), \tag{8}$$

where $\phi(\tau; \mu, \sigma^2) = \frac{1}{\sqrt{2\pi\sigma^2}} \exp\big(- \frac{(\tau-\mu)^2}{2\sigma^2}\big)$. Taking logarithms to both sides in Eq. 8, we get,

$$-\frac{(\tau - \mu_{off})^2}{2\sigma_{off}^2} - \log \sigma_{off} = -\frac{(\tau - \mu_{on})^2}{2\sigma_{on}^2} - \log \sigma_{on}, \tag{9}$$

When an intersection exists, the valid root $\tau$ between $\mu_{off}$ and $\mu_{on}$ in Eq. 9 is the separation point; Otherwise, it is set to the midpoint.

Given the separation point $\tau$, we identify candidate samples: offline points with unusually high scores ($d_i^{off} \geq \tau$) that behave online-like, and online points with unusually low scores ($d_i^{on} < \tau$) that behave offline-like. We then exchange the top-$M$ pairs between the offline batch and the online batch. Mathematically, we have

$$M = \min\Big(|\{d_i^{off} \geq \tau\}|, \ |\{d_i^{on} < \tau\}|, \ n\Big), \tag{10}$$

where $n$ is a hyperparameter setting the maximum number of pairs exchanged. This exchange leads to updated batches $b'_{off}$ and $b'_{on}$ that better reflect behavior consistency.

Built on the resulting batch $b' = b'_{off} \cup b'_{on}$, we enforce differentiated constraints during policy optimization. For offline-like samples in $b'_{off}$, we preserve the conservative objectives used in the original offline algorithm to ensure stability. For online-like samples in $b'_{on}$, we relax or remove such penalties, enabling the policy to adapt more flexibly to new behaviors.

## 5.4 THEORETICAL ANALYSIS

To understand why the proposed exchange mechanism improves off2on adaptation, we analyze its theoretical properties from: (i) **Geometric contraction**, showing that the proposed exchange scheme

monotonically reduces the distributional discrepancy between offline and online samples; and (ii) **Stability**, showing that the value estimation remains bounded throughout the exchange process.

Let $b_{off} = \{x_i^{\text{off}}\}_{i=1}^N$ and $b_{on} = \{x_j^{\text{on}}\}_{j=1}^N$ denote the offline and online batches. We study the distributional geometry of these batches by projecting them onto a one-dimensional statistic $d : \mathcal{X} \to \mathbb{R}$, yielding $\{d_i^{\text{off}}\}$ and $\{d_j^{\text{on}}\}$. The $H\Delta H$-divergence is defined as $d_{H\Delta H}(P, Q) = 2\sup_{(a,b)} |P\{d \in (a,b]\} - Q\{d \in (a,b]\}|$ (Ben-David et al., 2010).

**Theorem 1** (Monotone decay of threshold-class discrepancy). *Let $M$ be the number of accepted exchanges. The $H\Delta H$-divergence between the offline and online projections satisfies:*

$$d_{H\Delta H}^{(M)} \leq \max\left\{0,\ d_{H\Delta H}^{(0)} - \frac{2M}{N}\right\}. \tag{11}$$

Beyond geometric contraction, we further analyze whether such exchanges affect the stability of value estimation. Let $r_{\hat{Q}}(s,a) = (T^\pi \hat{Q})(s,a) - \hat{Q}(s,a)$ and $\mathcal{E}_\alpha = \alpha\mathbb{E}_{\mu_{\text{off}}}|r_{\hat{Q}}| + (1-\alpha)\mathbb{E}_{\mu_{\text{on}}}|r_{\hat{Q}}|$ as the mixed residual . This requires a mild coverage condition to relate residuals to value errors:

**Assumption 1** (Coverage and Control). *There exists $\kappa \geq 1$ such that for all bounded measurable $h$,*

$$\|h\|_\infty \leq \kappa\,\mathbb{E}_{\mu_{\text{mix}}}|h|, \qquad \mu_{\text{mix}} := \alpha\,\mu_{\text{off}} + (1-\alpha)\,\mu_{\text{on}}. \tag{12}$$

**Theorem 2** (Bounded value estimation error). *Assume $|r_{\hat{Q}}(s,a)| \leq B_T$ and the coverage condition in Assumption 1. Then after $M$ exchanges,*

$$\|\hat{Q} - Q^\pi\|_\infty \leq \frac{\kappa}{1-\gamma}\left(\mathcal{E}_\alpha^{(0)} + \frac{2B_T M}{N}\right). \tag{13}$$

These guarantees show that the exchange mechanism progressively aligns the offline and online distributions while keeping the value estimation bounded. Full proofs are provided in Appendix A.

# 6 INTEGRATING DARE INTO OFFLINE ALGORITHMS

We integrate DARE into two representative offline RL algorithms, Cal-QL and IQL, by applying differentiated updates to the exchanged batches $b'_{off}$ and $b'_{on}$. For offline-like samples in $b'_{off}$, we retain the original conservative objectives to preserve stability. For online-like samples in $b'_{on}$, we relax these constraints to facilitate adaptation.

## 6.1 DARE INSTANTIATIONS

**DARE-Cal-QL** We extend Cal-QL with DARE, resulting in the variant DARE-Cal-QL (DARE-C). Cal-QL applies a conservative regularizer $\mathcal{R}$ uniformly to all samples to mitigate value overestimation. In DARE-C, this penalty is applied selectively: $\mathcal{R}$ is retained for offline-like samples in $b'_{off}$ to preserve stability, but removed for online-like samples in $b'_{on}$ to facilitate adaptation. The original objective Eq. 1 becomes

$$\mathcal{L}_{\text{DARE}}^{\text{Cal-QL}}(Q) = \mathbb{E}_{(s,a,s')\sim\mathcal{D}}\Big[\mathbb{1}_{\{(s,a,s')\in b'_{off}\}}\big((Q(s,a) - \hat{\mathcal{B}}^\pi \hat{Q}_{\text{target}}(s,a))^2 + \alpha\mathcal{R}\big)$$
$$+ \mathbb{1}_{\{(s,a,s')\in b'_{on}\}}\big(Q(s,a) - \hat{\mathcal{B}}^\pi \hat{Q}_{\text{target}}(s,a)\big)^2\Big]. \tag{14}$$

**DARE-IQL** When applied to IQL, DARE brings in a variant DARE-IQL (DARE-I). IQL trains the Q-function via value regression and updates the policy through advantage-weighted regression. In DARE-I, these updates are differentiated: for offline-like samples $b'_{off}$, we retain the original value targets and policy regression; for online-like samples $b'_{on}$, we replace the value targets with TD-based maximum-Q estimates and switch the policy update to an entropy-regularized SAC objective (Haarnoja et al., 2018). The original Q-function and policy updates in and Eq. 3 and Eq. 4 become:

$$\mathcal{L}_{\text{DARE}}^{\text{IQL}}(Q) = \mathbb{E}_{(s,a,s')\sim\mathbf{D}}\Big[\mathbb{1}_{\{(s,a,s')\in b'_{\text{off}}\}}\big(r(s,a) + \gamma V(s') - Q(s,a)\big)^2$$
$$+ \mathbb{1}_{\{(s,a,s')\in b'_{\text{on}}\}}\big(r(s,a) + \gamma\max_{a'}\hat{Q}_{\text{target}}(s',a') - Q(s,a)\big)^2\Big], \tag{15}$$

and

$$\mathcal{L}_{\text{DARE}}^{\text{IQL}}(\pi) = \mathbb{E}_{(s,a)\sim\mathbf{D}}\left[\mathbb{1}_{\{(s,a)\in b'_{\text{off}}\}}\exp\left(\beta(\hat{Q}_{\text{target}}(s,a)-V(s))\right)\log\pi(a\mid s)\right]$$
$$+ \mathbb{E}_{s\sim\mathbf{D},\tilde{a}\sim\pi(\cdot\mid s)}\left[\mathbb{1}_{\{(s,\tilde{a})\in b'_{\text{on}}\}}\left(\alpha\log\pi(\tilde{a}\mid s)-Q(s,\tilde{a})\right)\right]. \tag{16}$$

## 6.2 ALGORITHM SUMMARY

Algorithm 1 summarizes DARE, intigrating the exchange mechanism with modified value and policy updates. We instantiate it on Cal-QL and IQL as representative baselines.

---

**Algorithm 1** Dynamic Alignment for Release (**DARE**)

---

1: **Initialize:** Offline policy $\pi_{off}$ as $\pi_{on}$ with Q networks, diffusion behavior model $\nu_0(\cdot\mid s)$.
2: **Initialize:** Replay buffers $\mathcal{D}_{off}$ with offline data, $\mathcal{D}_{on}$ with online data, energy function $f_\phi$.
3: **for** each iteration **do**
4:     Interact with environment using $\pi_{on}$ and collect new transitions to $\mathcal{D}_{on}$.
5:     Sample batch $b = b_{off}\cup b_{on} = \{(s_i,a_i,r_i,s_i')\}$ from $\mathcal{D}_{off}\cup\mathcal{D}_{on}$.
6:     Sample actions $\{\hat{a}_i\}$ via energy-guided sampling process under state $\{s_i\}$ via Eq. 6.
7:     Compute the alignment scores in Eq. 7 and find the intersection point $\tau$ by Eq. 9.
8:     Exchange $M$ samples between $b_{off}$ and $b_{on}$ by Eq. 10, then get $b'_{off}$ and $b'_{on}$.
9:     Update policy using $\mathcal{L}_{\text{Cal-QL}}$ in Eq. 14 or $\mathcal{L}_{\text{IQL}}$ in Eq. 15 and Eq. 16 with $b'_{off}$ and $b'_{on}$.
10:     Update energy function $f_\phi$.
11: **end for**

---

## 7 EXPERIMENTS

In this section, we present empirical evaluations of DARE against strong baselines in off2on RL. We begin by comparing DARE with representative offline RL algorithms on the D4RL benchmark. We then evaluate DARE along two dimensions: *Robustness* and *Stability*. To evaluate robustness, we sweep the exchange limit and measure sensitivity to excessive sample swaps. Stability is accessed by increasing the update-to-data (UTD) ratio and observing sensitivity to aggressive update schedules. We also compare our Gaussian-fitting strategy against a direct score-based exchange. In the end, the ablation studies are performed to demonstrate the contributions of the energy function and the exchange mechanism.

### 7.1 BENCHMARKS AND BASELINES

We evaluate DARE on two standard D4RL benchmarks (Fu et al., 2020): MuJoCo Locomotion and AntMaze Navigation, -both using the "-v2" version. As for baselines, in **CQL group**, we compare DARE-C against CQL (Kumar et al., 2020), Cal-QL, and EDIS-C (Liu et al., 2024), a Cal-QL variant; in **IQL group**, we compare DARE-I against IQL, PEX (Zhang et al., 2023), and EDIS-I, the IQL counterpart of EDIS-C. For fairness, EDIS-C and DARE-C in the CQL group are initialized from the same Cal-QL models, and all models in the IQL group are from the same IQL models. All above models are trained offline for 1M steps and fine-tuned online for 0.2M steps. We average the results over the last four evaluations and five random seeds. Additional implementation details, including the performance of the initial offline-trained models, are provided in Appendix B.

### 7.2 OVERALL PERFORMANCE

The fine-tuning results are presented under both CQL group and IQL group in Tab. 1. Overall, DARE brings a significant improvement of around $15\%$ total score of all datasets in both groups. It also consistently outperforms the baselines across different tasks, with highest scores on 11/13 tasks under the CQL group and on 10/13 tasks under the IQL group. These results demonstrate that the proposed exchange mechanism brings consistent benefits across diverse benchmarks in the off2on setting. Appendix C provides the complete fine-tuning learning curves, which illustrate how DARE maintains stable improvements across both locomotion and antmaze tasks.

Table 1: Performance after 0.2M online fine-tuning. Each result is averaged over the final 4 evaluations and 5 random seeds $\pm$ standard deviation. The "-C" and "-I" suffixes indicate the implementation based on Cal-QL and IQL, respectively. The highest scores are **bolded**.

| Dataset | CQL Group | | | | IQL Group | | | |
|---|---|---|---|---|---|---|---|---|
| | Base (CQL) | Cal-QL | EDIS-C | DARE-C | Base (IQL) | PEX | EDIS-I | DARE-I |
| HC-ME | 96.3±1.6 | 96.4±0.9 | 95.1±0.7 | **97.0±0.8** | 91.7±2.3 | 89.2±3.6 | 91.4±3.9 | 93.5±2.1 |
| H-ME | 111.9±0.9 | 111.9±0.7 | 111.9±1.7 | 112.0±0.4 | 53.9±39.0 | 90.2±20.1 | 99.7±13.5 | 102.3±9.6 |
| W2D-ME | 110.3±0.5 | 110.4±0.5 | 108.2±7.1 | **111.2±0.6** | 111.8±4.9 | 114.8±3.0 | 113.1±0.9 | 114.7±1.6 |
| HC-MR | 50.9±0.5 | 51.1±1.1 | 56.4±2.8 | **78.5±1.4** | 47.4±1.0 | 53.3±1.2 | 45.7±0.7 | 49.4±1.0 |
| H-MR | 82.1±33.2 | 93.0±13.4 | 100.9±5.9 | 103.5±1.1 | 87.0±28.2 | 93.5±13.7 | 93.7±9.7 | 97.8±3.0 |
| W2D-MR | 86.9±3.4 | 88.4±4.6 | 108.9±4.2 | 110.6±1.9 | 91.8±6.2 | 92.0±6.4 | 89.2±3.8 | 97.0±3.1 |
| HC-M | 64.6±2.6 | 66.9±1.8 | 68.8±1.8 | **79.3±3.2** | 57.8±1.3 | 65.8±2.9 | 49.3±0.3 | 63.3±1.8 |
| H-M | 81.9±8.1 | 86.6±9.1 | 94.7±7.2 | **99.6±4.7** | 77.5±22.2 | 84.3±20.1 | 58.9±6.3 | 99.8±2.8 |
| W2D-M | 83.0±0.7 | 83.4±1.5 | 85.9±1.5 | **87.0±3.4** | 85.8±7.6 | 90.2±13.1 | 86.4±1.6 | 92.2±3.2 |
| total (L) | 767.9 | 788.1 | 830.8 | **878.7** | 704.7 | 773.3 | 727.4 | **810.0** |
| AM-MD | 85.8±5.6 | 86.8±4.7 | 93.4±2.9 | **94.8±4.0** | 82.0±6.0 | 82.8±6.4 | 84.9±5.8 | 78.6±5.4 |
| AM-MP | 86.2±4.4 | 89.1±5.4 | 94.4±2.9 | 93.4±4.2 | 80.4±3.6 | 81.7±4.7 | 78.1±7.2 | 83.5±4.8 |
| AM-UD | 82.4±7.4 | 90.1±7.8 | 86.1±3.7 | **91.1±6.5** | 31.6±16.1 | 4.6±8.4 | 34.9±9.3 | 74.4±11.7 |
| AM-U | 93.2±1.8 | 95.5±1.3 | 95.2±2.9 | **97.7±2.5** | 90.5±3.5 | 92.9±3.7 | 92.2±2.3 | 94.0±3.3 |
| total (AM) | 347.6 | 361.5 | 369.1 | **377.0** | 284.5 | 262.0 | 290.1 | **330.5** |
| total | 1115.5 | 1149.6 | 1200.1 | **1255.7** | 989.2 | 1035.3 | 1017.5 | **1140.5** |

*Abbreviations:* HC = `halfcheetah`, H = `hopper`, W2D = `walker2d`, L = locomotion, AM = antmaze; M = medium, ME = medium-expert, MR = medium-replay, MD = medium-diverse, MP = medium-play, UD = umaze-diverse, U = umaze. All environments use `-v2` version.

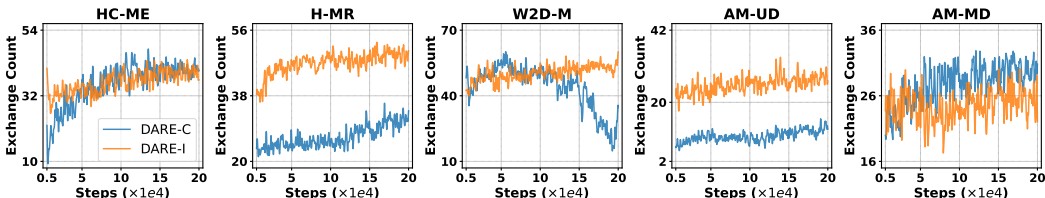

Figure 2: Adaptive exchange counts during online training across different tasks.

## 7.3 IMPACT OF EXCHANGE CAP

Since DARE relies on the exchange mechanism to reassign samples between offline and online batches, its effectiveness depends on the accuracy of the estimated intersection point. Unreliable exchanges will reduce training efficiency or even destabilize learning.

To mitigate potential instability from excessive exchanges, we introduce the hyperparameter $n$ in Eq. 10 to cap the number of swaps per batch. Fig. 2 shows that DARE adapts exchange frequency differently across environments, with the actual number of exchanges in most cases remaining relatively low (typically $< 70$). We then compare capped ($n = 32$) and uncapped ($n = \infty$) settings on MuJoCo Locomotion tasks in Tab. 2. Under DARE-C, $n = 32$ achieves better overall performance, suggesting that excessive swaps cause instability. In contrast, DARE-I obtains higher scores without such a limit. Such observation indicates that the smaller $n$ restricts aggressive swapping and the larger one allow more flexible adaptation. A comprehensive breakdown of exchange dynamics across environments, together with detailed results under different $n$, is provided in Appendix D.

| | Dataset | n = 32 | n = ∞ |
|---|---|---|---|
| DARE-C | M | **265.9** | 258.5 |
| | MR | **292.6** | 283.9 |
| | ME | 320.2 | 320.3 |
| | Total | **878.7** | 862.7 |
| DARE-I | M | 249.7 | **255.3** |
| | MR | 240.8 | **244.2** |
| | ME | 300.3 | **310.5** |
| | Total | 790.8 | **810.0** |

Table 2: Effect of the maximum exchange limit $n$ for DARE-C and DARE-I.

## 7.4 IMPACT OF LARGE UPDATE-TO-DATA RATIOS

To assess the stability of DARE, we investigate the effect of the UTD ratio, which determines the number of gradient updates performed per environment step.

While a higher ratio accelerates learning, it also induces more frequent exchange between offline and online-like samples, potentially affecting stability. We evaluate DARE's performance with an aggressive setting with UTD = 10 after 0.1M fine-tuning steps on MuJoCo locomotion tasks. As shown in Tab. 3, DARE-I achieves the highest total scores under UTD = 10, compared to both PEX and EDIS-I. These results indicate that DARE remains stable even when the exchange becomes more frequent under aggressive update regimes. Detailed results of the UTD experiments, along with the corresponding learning curves, are provided in Appendix E.

| Dataset | EDIS-I | PEX | DARE-I |
|---------|--------|-----|--------|
| HC-MR | 45.0±0.9 | **58.4±2.8** | 49.7±0.5 |
| H-MR | 94.1±9.7 | 88.5±22.8 | **100.2±2.4** |
| W2D-MR | 75.9±9.0 | 99.9±10.5 | **101.4±6.7** |
| HC-M | 48.8±0.2 | **76.1±1.6** | 66.3±1.4 |
| H-M | 60.8±5.3 | 84.6±21.4 | **101.2±4.3** |
| W2D-M | 81.4±5.5 | 92.5±18.2 | **99.1±1.6** |
| total | 406.0 | 495.8 | **517.9** |

Table 3: Performance of IQL-based methods under UTD = 10.

### 7.5 EFFECTIVENESS OF GAUSSIAN FITTING

To evaluate the design choice of using Gaussian fitting by Eq. 8, we compare it with a direct exchange. The direct exchange sorts the sampled training batch $b$ according to the alignment scores in Eq. 7, and assigns the top-ranked samples to $b'_{off}$ and the remaining to $b'_{on}$. Fig. 3 depicts the learning curves under the Cal-QL framework. In most environments, our Gaussian-fitting strategy achieves more stable performance compared to the

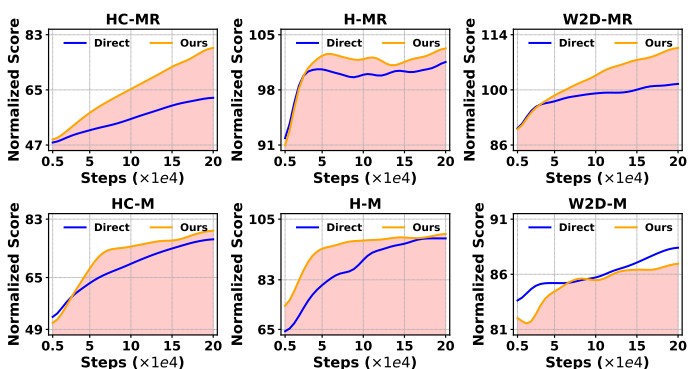

Figure 3: Comparison of Gaussian fitting and direct exchange.

direct exchange. Although direct sorting is simple, its cutoff depends heavily on the sampled batch and can fluctuate. In contrast, Gaussian fitting models the score distributions and uses their intersection point as the decision boundary, which will make the partition more stable.

### 7.6 ABLATION STUDY

In the end, we perform an ablation study to assess the contribution of each component in DARE-I. As shown in Fig. 4, removing either the energy guidance in the diffusion model or the sample exchange mechanism leads to significant performance drops across multiple tasks. These results highlight the important roles of both components in ensuring the overall effectiveness of DARE-I. Additional ablation results are provided in Appendix F due to space constraints.

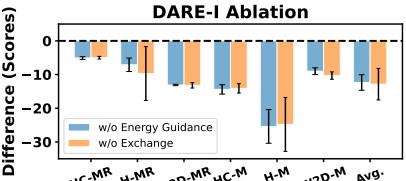

Figure 4: Ablation study of DARE-I.

## 8 CONCLUSION

We have presented DARE, a distribution-aware framework for offline-to-online reinforcement learning that adaptively relaxes constraints at the sample level. By integrating energy-guided diffusion, alignment scoring, Gaussian fitting, and an exchange mechanism, DARE distinguishes offline-like from online-like samples, preserving conservative constraints on the former while adaptively relaxing them for the latter. Our analysis establishes that DARE reduces the offline-to-online distribution discrepancy while ensuring bounded value estimation. Experiments on MuJoCo and AntMaze benchmarks demonstrate that integrating DARE into existing methods consistently improves stability during fine-tuning, robustness, and overall policy performance. Future directions include exploring richer behavior models to better capture multi-modal offline data and more effectively guide the exchange process.

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

APPENDIX

**Ethics Statement** This work does not involve human subjects, personally identifiable information, or sensitive user data. All datasets used are publicly available benchmark datasets. Our methods focus on algorithmic improvements for off2on RL and do not raise foreseeable risks of harmful societal or environmental impacts.

**Reproducibility Statement** We have made every effort to ensure reproducibility of our results. A full description of the proposed algorithm, training settings is provided in the main text. Hyperparameters used are documented in the appendix. Source code and scripts for reproducing all experiments are included in the supplementary materials.

**LLM Usage** Large language models (LLMs) are used solely as writing assistants to help polish the manuscript's grammar and clarity.

## A PROOFS

### A.1 SETUP AND NOTATION

Let $b_{off} = \{x_i^{\text{off}}\}_{i=1}^N$ and $b_{on} = \{x_j^{\text{on}}\}_{j=1}^N$ be the offline and online batches (cardinality $N$ each). Let $d : \mathcal{X} \to \mathbb{R}$ be a one-dimensional statistic. Write $d_i^{\text{off}} := d(x_i^{\text{off}})$ and $d_j^{\text{on}} := d(x_j^{\text{on}})$. Define the empirical distributions on the projected values as

$$\mu_{\text{off}} := \frac{1}{N} \sum_{i=1}^N \delta_{x_i^{\text{off}}}, \qquad \mu_{\text{on}} := \frac{1}{N} \sum_{j=1}^N \delta_{x_j^{\text{on}}}, \tag{17}$$

where $\delta_x$ denotes the Dirac measure at $x$, characterized by the property

$$\int f(y) \, d\delta_x(y) = f(x), \qquad \forall f : \mathcal{X} \to \mathbb{R}. \tag{18}$$

Accordingly, for any test function $f$,

$$\mathbb{E}_{\mu_{\text{off}}}[f] = \int f \, d\mu_{\text{off}} = \frac{1}{N} \sum_{i=1}^N f(x_i^{\text{off}}), \qquad \mathbb{E}_{\mu_{\text{on}}}[f] = \frac{1}{N} \sum_{j=1}^N f(x_j^{\text{on}}). \tag{19}$$

That is, expectations under $\mu_{\text{off}}$ and $\mu_{\text{on}}$ coincide exactly with sample averages over the corresponding batches.

Let $F_{\text{off}}(t)$ and $F_{\text{on}}(t)$ be the empirical CDFs (Cumulative Distribution Function) of $d$ under the two measures and $\Delta F(t) := F_{\text{off}}(t) - F_{\text{on}}(t)$. Define $\Delta F_{\max} := \sup_t \Delta F(t)$, $\Delta F_{\min} := \inf_t \Delta F(t)$, and the CDF amplitude $A := \Delta F_{\max} - \Delta F_{\min} \geq 0$.

Consider the threshold-interval hypothesis class $\mathcal{I}_{\text{thr}} = \{(a,b] \subset \mathbb{R} : a < b\}$. Following Ben-David et al. (2010), the $H\Delta H$-divergence is used between the $d$-projections of two measures $P, Q$ by

$$d_{H\Delta H}(P, Q) := 2 \sup_{(a,b] \in \mathcal{I}_{\text{thr}}} \big| P\{d \in (a,b]\} - Q\{d \in (a,b]\} \big|. \tag{20}$$

Since $P\{d \in (a,b]\} = F_P(b) - F_P(a)$, one has

$$d_{H\Delta H}(P, Q) = 2\big(\Delta F_{\max} - \Delta F_{\min}\big) = 2A. \tag{21}$$

**Symmetric exchange.** An *exchange* replaces one offline atom $x$ (with value $d_x$) by an online atom $y$ (with value $d_y$) symmetrically: $x : \text{off} \to \text{on}$ and $y : \text{on} \to \text{off}$. For all $t \in \mathbb{R}$, the updated difference of CDFs satisfies

$$\Delta F'(t) = \Delta F(t) - \frac{1}{N} \mathbf{1}\{t \geq d_x\} + \frac{1}{N} \mathbf{1}\{t \geq d_y\}. \tag{22}$$

An exchange is *effective* if it reduces the amplitude by at least one unit of granularity, i.e., $A' \leq A - \frac{1}{N}$, where $A'$ is the post-exchange amplitude. Let $M$ be the number of accepted (effective) exchanges.

## A.2 Existence of effective pairs and capacity bound

Let $t_{\max} \in \arg\max_t \Delta F(t)$ and $t_{\min} \in \arg\min_t \Delta F(t)$. Define

$$S_{\mathrm{off}}^R := \{i : d_i^{\mathrm{off}} \geq t_{\max}\}, \qquad S_{\mathrm{on}}^L := \{j : d_j^{\mathrm{on}} \leq t_{\min}\}. \tag{23}$$

**Lemma 1** (Sufficient condition for an effective exchange). *If $S_{\mathrm{off}}^R \neq \emptyset$ and $S_{\mathrm{on}}^L \neq \emptyset$, any pair $(x, y)$ with $x$ chosen from $S_{\mathrm{off}}^R$ and $y$ from $S_{\mathrm{on}}^L$ induces an exchange that reduces the amplitude by at least $1/N$, i.e. $A' \leq A - \frac{1}{N}$.*

*Proof.* Pick $x$ with $d_x \geq t_{\max}$ and $y$ with $d_y \leq t_{\min}$. By Eq. 22, for $t = t_{\max}$ we have $\mathbf{1}\{t_{\max} \geq d_x\} = 1$ (since $d_x \leq t_{\max}$ is false but $\Delta F$'s jump locations are right-closed; one may take the immediate right limit), and $\mathbf{1}\{t_{\max} \geq d_y\} = 1$ because $d_y \leq t_{\min} \leq t_{\max}$. Thus $\Delta F'(t_{\max}) = \Delta F(t_{\max}) - 1/N + 1/N = \Delta F(t_{\max})$ or, with standard right/left-limit care, it cannot increase beyond $\Delta F_{\max}$. At $t = t_{\min}$, we get $\Delta F'(t_{\min}) = \Delta F(t_{\min}) - 1/N + 1/N = \Delta F(t_{\min})$ or it cannot decrease below $\Delta F_{\min}$.

Crucially, for any $t > t_{\max}$ we have $\Delta F'(t) = \Delta F(t) - 1/N + 1/N = \Delta F(t)$, while for any $t \in (t_{\min}, t_{\max})$, $\Delta F'(t) = \Delta F(t) - 1/N + 1/N = \Delta F(t)$. The only strict change that impacts the amplitude occurs on $(-\infty, t_{\min}]$ and on $[t_{\max}, \infty)$ through the step structure: moving an offline atom from the rightmost side suppresses the upper envelope by $1/N$ and moving an online atom from the leftmost side lifts the lower envelope by $1/N$. Therefore $A' \leq A - 1/N$. $\qquad\square$

**Proposition 1** (Capacity bound). *Let $A_0$ be the initial amplitude. Any sequence of accepted exchanges satisfies $M \leq \lfloor A_0 N \rfloor$.*

*Proof.* Each accepted exchange decreases $A$ by at least $1/N$ and $A \geq 0$ always. Hence after $M$ steps $A_M \leq A_0 - M/N \geq 0$, which implies $M \leq A_0 N$. $\qquad\square$

## A.3 Monotone decrease of threshold-class discrepancy

**Theorem 3** (Monotone decay of threshold-class discrepancy on the projected $d$-axis). *Let $M$ be the number of accepted exchanges (as per the rule $A' \leq A - \frac{1}{N}$). Then for the threshold-interval class,*

$$d_{H\Delta H}^{(M)} \leq \max\left\{0, \, d_{H\Delta H}^{(0)} - \frac{2M}{N}\right\}. \tag{24}$$

*Proof.* Let $A_k$ denote the amplitude after $k$ accepted exchanges. By definition of an effective exchange, each step decreases the amplitude by at least $1/N$:

$$A_k \leq A_{k-1} - \frac{1}{N}, \qquad k = 1, 2, \ldots, M. \tag{25}$$

Iterating this inequality yields $A_M \leq \max\{0, A_0 - M/N\}$. Finally, since $d_{H\Delta H} = 2A$ by Eq. 21, we obtain the desired bound. $\qquad\square$

**Remark 1.** Theorem 3 guarantees that the discrepancy under the threshold-interval hypothesis class decreases monotonically along the $d$-projection. In other words, the one-dimensional separability measured by $d_{H\Delta H}$ shrinks deterministically with each accepted exchange. However, this control is specific to the projected $d$-axis and the threshold function class. It does not imply that the total variation distance between $\mu_{\mathrm{off}}$ and $\mu_{\mathrm{on}}$ over the full $(s, a)$-space decreases. In the following section we provide an independent stability analysis in the original $(s, a)$ domain.

## A.4 Stability of training residual

Let $r_{\hat{Q}}(s, a) = (T^\pi \hat{Q})(s, a) - \hat{Q}(s, a)$ and assume a uniform bound

$$|r_{\hat{Q}}(s, a)| \leq B_T \quad \text{for all } (s, a). \tag{26}$$

Fix $\alpha \in (0, 1)$ and define the mixed residual

$$\mathcal{E}_\alpha(\hat{Q}; \mu_{\mathrm{off}}, \mu_{\mathrm{on}}) := \alpha \, \mathbb{E}_{\mu_{\mathrm{off}}} |r_{\hat{Q}}| + (1 - \alpha) \, \mathbb{E}_{\mu_{\mathrm{on}}} |r_{\hat{Q}}|. \tag{27}$$

**Lemma 2** (One-step stability). *One symmetric exchange changes $\mathcal{E}_\alpha$ by at most $2B_T/N$ in absolute value:*

$$\left| \mathcal{E}_\alpha(\hat{Q}; \mu'_{\text{off}}, \mu'_{\text{on}}) - \mathcal{E}_\alpha(\hat{Q}; \mu_{\text{off}}, \mu_{\text{on}}) \right| \leq \frac{2B_T}{N}. \tag{28}$$

*Proof.* Let $h(s, a) := |r_{\hat{Q}}(s, a)|$, so by Eq. 26 we have $\|h\|_\infty \leq B_T$. Using the standard total-variation inequality $|\mathbb{E}_\mu h - \mathbb{E}_\nu h| \leq 2\|h\|_\infty \operatorname{TV}(\mu, \nu)$ and the fact that a single-atom replacement in a uniform empirical measure has $\operatorname{TV}(\mu', \mu) = 1/N$, the claimed bounds for the offline and online parts follow

$$\left| \alpha(\mathbb{E}_{\mu'_{\text{off}}}|r| - \mathbb{E}_{\mu_{\text{off}}}|r|) \right| \leq \alpha \cdot 2B_T \cdot \frac{1}{N},$$

$$\left| (1-\alpha)(\mathbb{E}_{\mu'_{\text{on}}}|r| - \mathbb{E}_{\mu_{\text{on}}}|r|) \right| \leq (1-\alpha) \cdot 2B_T \cdot \frac{1}{N}. \tag{29}$$

Summing the two bounds establishes the desired inequality. $\square$

**Proposition 2** (Accumulated stability). *After $M$ (possibly reverted-or-accepted, but realized) exchanges,*

$$\left| \mathcal{E}_\alpha^{(M)} - \mathcal{E}_\alpha^{(0)} \right| \leq \frac{2B_T M}{N}. \tag{30}$$

*Proof.* By Lemma 2, each exchange changes $\mathcal{E}_\alpha$ by at most $2B_T/N$ in absolute value. Summing these deviations over $M$ steps yields the bound above. $\square$

**Assumption 2** (Coverage and Control). *There exists $\kappa \geq 1$ such that for all bounded measurable $h$,*

$$\|h\|_\infty \leq \kappa \, \mathbb{E}_{\mu_{\text{mix}}}|h|, \qquad \mu_{\text{mix}} := \alpha \, \mu_{\text{off}} + (1-\alpha) \, \mu_{\text{on}}. \tag{31}$$

**Proposition 3** (Bellman contraction). $\|\hat{Q} - Q^\pi\|_\infty \leq \frac{1}{1-\gamma} \|r_{\hat{Q}}\|_\infty$.

**Theorem 4** (From residual to $\infty$-norm error). *Under Assumption 2 and Eq. 26,*

$$\|\hat{Q} - Q^\pi\|_\infty \leq \frac{\kappa}{1-\gamma} \mathcal{E}_\alpha(\hat{Q}; \mu_{\text{off}}, \mu_{\text{on}}). \tag{32}$$

*Consequently, after $M$ exchanges,*

$$\|\hat{Q} - Q^\pi\|_\infty \leq \frac{\kappa}{1-\gamma} \left( \mathcal{E}_\alpha^{(0)} + \frac{2B_T M}{N} \right). \tag{33}$$

*Proof.* By Proposition 3 and Assumption 2 with $h = r_{\hat{Q}}$, $\|\hat{Q} - Q^\pi\|_\infty \leq \frac{1}{1-\gamma}\|r_{\hat{Q}}\|_\infty \leq \frac{\kappa}{1-\gamma} \mathbb{E}_{\mu_{\text{mix}}}|r_{\hat{Q}}| = \frac{\kappa}{1-\gamma} \mathcal{E}_\alpha$. The second inequality follows from Proposition 2. $\square$

**Remark 2.** Assumption 1 requires that the mixed sampling distribution $\mu_{\text{mix}} = \alpha \mu_{\text{off}} + (1 - \alpha)\mu_{\text{on}}$ provides sufficient coverage of the state–action space so that the worst-case residual can be controlled by its average under $\mu_{\text{mix}}$. In particular, the constant $\kappa$ quantifies the potential mismatch: $\kappa = 1$ corresponds to ideal coverage, whereas larger values of $\kappa$ indicate that the distribution may under-sample certain regions, making the sup-norm error bound looser.

### A.5 SUMMARY OF GUARANTEES

- **Geometry on $d$ (Theorem 3).** Accepted exchanges monotonically reduce the threshold-class discrepancy: $d_{H \Delta H}^{(M)} \leq \max\{0, d_{H \Delta H}^{(0)} - 2M/N\}$, with capacity $M \leq \lfloor A_0 N \rfloor$.
- **Stability in $(s, a)$ (Theorem 4).** Regardless of geometry on $d$, the mixed residual and the $\infty$-norm value error remain controlled: $|\mathcal{E}_\alpha^{(M)} - \mathcal{E}_\alpha^{(0)}| \leq 2B_T M/N$ and $\|\hat{Q} - Q^\pi\|_\infty \leq \frac{\kappa}{1-\gamma}(\mathcal{E}_\alpha^{(0)} + 2B_T M/N)$.

## B EXPERIMENTAL DETAILS

In our experiments, we evaluate DARE on two standard D4RL benchmarks (Fu et al., 2020): Mu-JoCo Locomotion and AntMaze Navigation. For instantiation, we extend Cal-QL (Nakamoto et al., 2023) to DARE-C and IQL (Kostrikov et al., 2021) to DARE-I. In the CQL-based group, we compare against CQL (Kumar et al., 2020), Cal-QL, and EDIS-C (Liu et al., 2024), a Cal-QL variant. In the IQL-based group, we include IQL, PEX (Zhang et al., 2023), and EDIS-I, the IQL counterpart of EDIS-C. All models are first trained offline for 1M steps and then fine-tuned online for 0.2M steps. The results are averaged over the last four evaluations and five random seeds. For a fair comparison, **all methods are initialized from the same offline-trained models, using Cal-QL models for EDIS-C and DARE-C and IQL models for PEX, EDIS-I, and DARE-I.**

### B.1 HYPERPARAMETERS FOR DARE

For the implementation of DARE, there is only one additional hyperparameter, the maximum exchange number $n$. We set $n = 32$ for DARE-C and $n = \infty$ for DARE-I. In addition, for the SAC-style policy update in DARE-I, the entropy coefficient $\alpha$ is set to 0.2 on locomotion tasks and 0.01 on AntMaze tasks, respectively.

### B.2 HYPERPARAMETERS FOR CQL AND CAL-QL

We implement the CQL and Cal-QL based on `https://github.com/tinkoff-ai/CORL`, and primarily follow the authors' recommended hyperparameters (Tarasov et al., 2023). Since CQL-based algorithms are known to be sensitive to hyperparameter choices, we provide the exact settings in our experiments to facilitate the reproducibility. Please refer to Tab. 4 for the details about the hyperparameters in our CQL-based implementation.

Table 4: Hyperparameters in CQL-based implementation.

| Hyperparameter | Mojoco locomotion | AntMaze navigation |
|---|---|---|
| **General Settings** | | |
| Replay buffer size | 2,000,000 | 2,000,000 |
| Batch size | 256 | 256 |
| Discount factor $\gamma$ | 0.99 | 0.99 |
| Reward scale / bias | 1.0 / 0.0 | 10.0 / -5.0 |
| Normalize states | True | False |
| Normalize reward | False | True |
| Orthogonal initialization | True | True |
| Is sparse reward | False | True |
| **CQL Hyperparameters** | | |
| Policy learning rate | $1 \times 10^{-4}$ | $1 \times 10^{-4}$ |
| Critic learning rate | $3 \times 10^{-4}$ | $3 \times 10^{-4}$ |
| Soft target update rate $\tau$ | 0.005 | 0.005 |
| Target update period | 1 | 1 |
| Automatic entropy tuning | True | True |
| Backup entropy | False | False |
| CQL regularization ($\alpha$, offline / online) | 10.0 / 10.0 | 5.0 / 5.0 |
| CQL Lagrange | False | True |
| CQL temperature | 1.0 | 1.0 |
| Target action gap | $-1.0$ | 0.8 |
| Max target backup | False | True |
| Clip diff range | $[-200, \infty)$ | $[-200, \infty)$ |
| Importance sampling | True | True |
| **Network Architecture** | | |
| Q-network hidden layers | 3 | 5 |
| Hidden dimension (actor / critic) | 256 / 256 | 256 / 256 |

## B.3 HYPERPARAMETERS FOR IQL AND PEX

We implement IQL and PEX based on `https://github.com/Haichao-Zhang/PEX`, the hyperparameters of which is illustrated in Tab. 5.

Table 5: Hyperparameters for the IQL-based experiments.

| Hyperparameter | Value |
|---|---|
| Discount factor $\gamma$ | 0.99 |
| Hidden dimension | 256 |
| Number of hidden layers | 2 |
| Batch size | 256 |
| Learning rate | $3 \times 10^{-4}$ |
| Target update rate | 0.005 |
| Expectile parameter $\tau$ | 0.9, AntMaze / 0.7, Locomotion |
| Inverse temperature $\beta$ | 10.0, AntMaze / 3.0, Locomotion |

## B.4 HYPERPARAMETERS FOR EDIS

The implementation of EDIS are referred to `https://github.com/liuxhym/EDIS`. For EDIS-C, we use its official implementation of Cal-QL. For EDIS-I, we modify the classes of Q-function, value function, and policy function to match those in IQL and PEX, so that the same offline models can be loaded for initialization. The hyperparameters used in the EDIS module remain unchanged and, for convenience, are detailed in Tab. 6.

Table 6: Hyperparameters in EDIS.

| Hyperparameter | Value |
|---|---|
| Network Type (Denoising) | Residual MLP |
| Denoising Network Depth | 6 layers |
| Denoising Steps | 128 steps |
| Denoising Network Learning Rate | $3 \times 10^{-4}$ |
| Denoising Network Hidden Dimension | 1024 units |
| Denoising Network Batch Size | 256 samples |
| Denoising Network Activation Function | ReLU |
| Denoising Network Optimizer | Adam |
| Learning Rate Schedule (Denoising Network) | Cosine Annealing |
| Training Epochs (Denoising Network) | 50,000 epochs |
| Training Interval Environment Step (Denoising Network) | Every 10,000 steps |
| Energy Network Hidden Dimension | 256 units |
| Negative Samples (Energy Network Training) | 10 |
| Energy Network Learning Rate | $1 \times 10^{-3}$ |
| Energy Network Activation Function | ReLU |
| Energy Network Optimizer | Adam |

## B.5 HYPERPARAMETERS FOR ENERGY-GUIDED DIFFUSION MODEL

The implementation of the energy-guided diffusion model is refered to `https://github.com/thu-ml/CEP-energy-guided-diffusion`. Briefly, the behavior model follows the architecture and training strategy of Chen et al. (2022). We train the models for $6 \times 10^5$ gradient steps using the Adam optimizer with a learning rate of $1 \times 10^{-4}$ and a batch size of 4096.

The energy guidance model $f_\phi$ is implemented as a 4-layer MLP with 256 hidden units and SiLU activations. Training is performed using the Adam optimizer with a learning rate of $3 \times 10^{-4}$ and a batch size of 256. To train this energy network, a contrastive learning objective based on self-

normalized energy labels is defined as:

$$\min_{\phi} \; \mathbb{E}_{p(t)} \, \mathbb{E}_{q_0(x_0^{(1:K)})} \, \mathbb{E}_{p(\epsilon^{(1:K)})} \left[ -\sum_{i=1}^{K} \frac{e^{-\beta \mathcal{E}(x_0^{(i)})}}{\sum_{j=1}^{K} e^{-\beta \mathcal{E}(x_0^{(j)})}} \log \frac{e^{-f_\phi(x_t^{(i)},t)}}{\sum_{j=1}^{K} e^{-f_\phi(x_t^{(j)},t)}} \right]. \tag{34}$$

Here, each perturbed sample is generated by the forward diffusion process: $x_t^{(i)} = \alpha_t x_0^{(i)} + \sigma_t \epsilon^{(i)}$, with constants $\alpha_t, \sigma_t$ defined by the diffusion schedule. The guidance scale $s$ follows the empirical settings from prior work (Lu et al., 2023a), as summarized in Tab. 7.

Table 7: Guidance scale $s$ across different environments.

| Dataset | Guidance Scale $s$ |
|---|---|
| walker2d-medium-expert-v2 | 5.0 |
| halfcheetah-medium-expert-v2 | 3.0 |
| hopper-medium-expert-v2 | 2.0 |
| walker2d-medium-replay-v2 | 5.0 |
| halfcheetah-medium-replay-v2 | 8.0 |
| hopper-medium-replay-v2 | 3.0 |
| walker2d-medium-v2 | 10.0 |
| halfcheetah-medium-v2 | 10.0 |
| hopper-medium-v2 | 8.0 |
| antmaze-umaze-v2 | 3.0 |
| antmaze-medium-play-v2 | 4.0 |
| antmaze-umaze-diverse-v2 | 1.0 |
| antmaze-medium-diverse-v2 | 3.0 |

### B.6 OFFLINE MODEL PERFORMANCE

Before fine-tuning, we present the performance of the offline-trained models on both Locomotion and AntMaze benchmarks. These results serve as the initialization for all compared methods, ensuring that performance differences in the online phase come solely from the fine-tuning strategy rather than the quality of the offline implementation codebases. Tab. 8 summarizes the scores of CQL, Cal-QL and IQL across different datasets.

Table 8: Offline training performance before online fine-tuning.

| Dataset | CQL | Cal-QL | IQL |
|---|---|---|---|
| HC-ME | 90.4±3.5 | 82.9±4.1 | 90.2±1.4 |
| H-ME | 108.1±2.4 | 109.0±2.1 | 32.8±32.6 |
| W2D-ME | 109.7±0.2 | 108.6±0.8 | 108.3±2.0 |
| HC-MR | 45.5±0.2 | 46.5±0.2 | 43.9±0.4 |
| H-MR | 90.2±10.5 | 64.8±27.1 | 84.2±13.6 |
| W2D-MR | 78.6±4.5 | 85.0±2.9 | 70.8±4.1 |
| HC-M | 47.1±0.3 | 48.3±0.3 | 48.1±0.1 |
| H-M | 63.9±1.5 | 61.9±2.0 | 56.2±4.4 |
| W2D-M | 81.9±1.2 | 83.7±1.2 | 74.0±3.1 |
| total (L) | 715.4 | 690.7 | 608.5 |
| AM-MD | 57.6±3.5 | 66.2±6.1 | 70.4±5.2 |
| AM-MP | 64.8±6.1 | 68.4±3.8 | 74.6±4.7 |
| AM-UD | 28.2±23.9 | 46.0±20.1 | 60.2±6.4 |
| AM-U | 87.6±4.6 | 76.6±2.7 | 91.4±0.8 |
| total (AM) | 238.2 | 257.2 | 296.6 |
| total | 953.6 | 947.9 | 905.1 |

# C    COMPARISONS OF ONLINE FINE-TUNING PROCESSES

## C.1    ONLINE FINE-TUNING PROCESSES FOR DARE-C

As a complement to the main results in Tab. 1, we depict the online training dynamics across algorithms. As shown in Figs. 5 and 6, our method consistently outperforms both Cal-QL and EDIS on the locomotion and AntMaze benchmarks, while exhibiting smoother training trajectories.

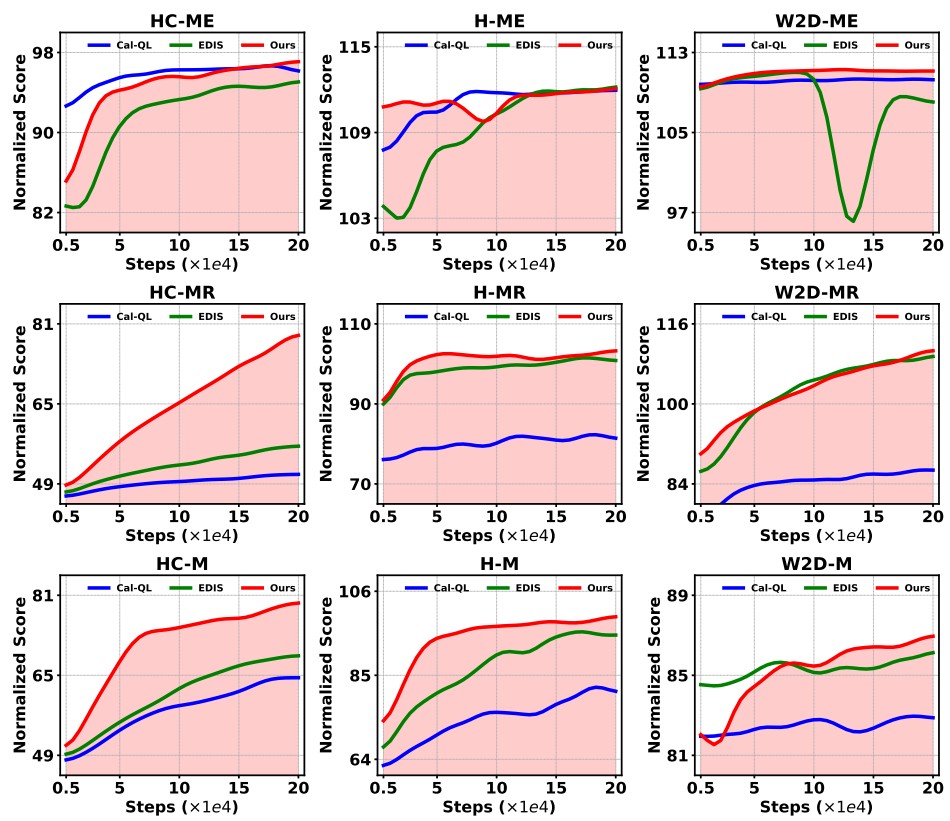

Figure 5: Online training curves on Locomotion benchmarks comparing Cal-QL, EDIS, and Ours.

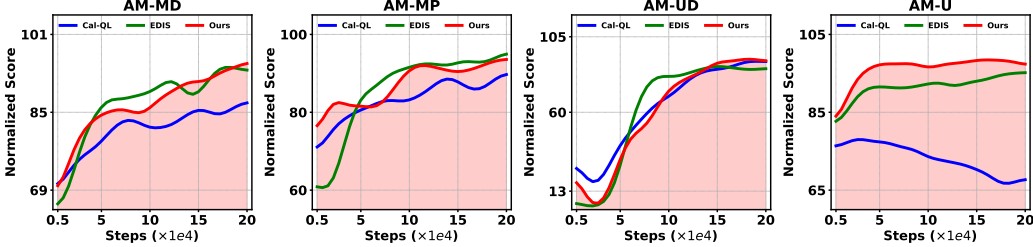

Figure 6: Online training curves on AntMaze benchmarks comparing Cal-QL, EDIS, and Ours.

## C.2    ONLINE FINE-TUNING PROCESSES FOR DARE-I

We also visualize the complete online performance under the IQL framework across all tasks. As shown in Figs. 7 and 8, our method improves over both PEX and EDIS in terms of learning dynamics and asymptotic returns, with the exceptions of *halfcheetah-medium*, *halfcheetah-medium-replay* and *antmaze-medium-diverse*. In general, our approach converges more smoothly and reaches higher normalized scores with fewer training steps, demonstrating better sample efficiency and robustness during online fine-tuning.

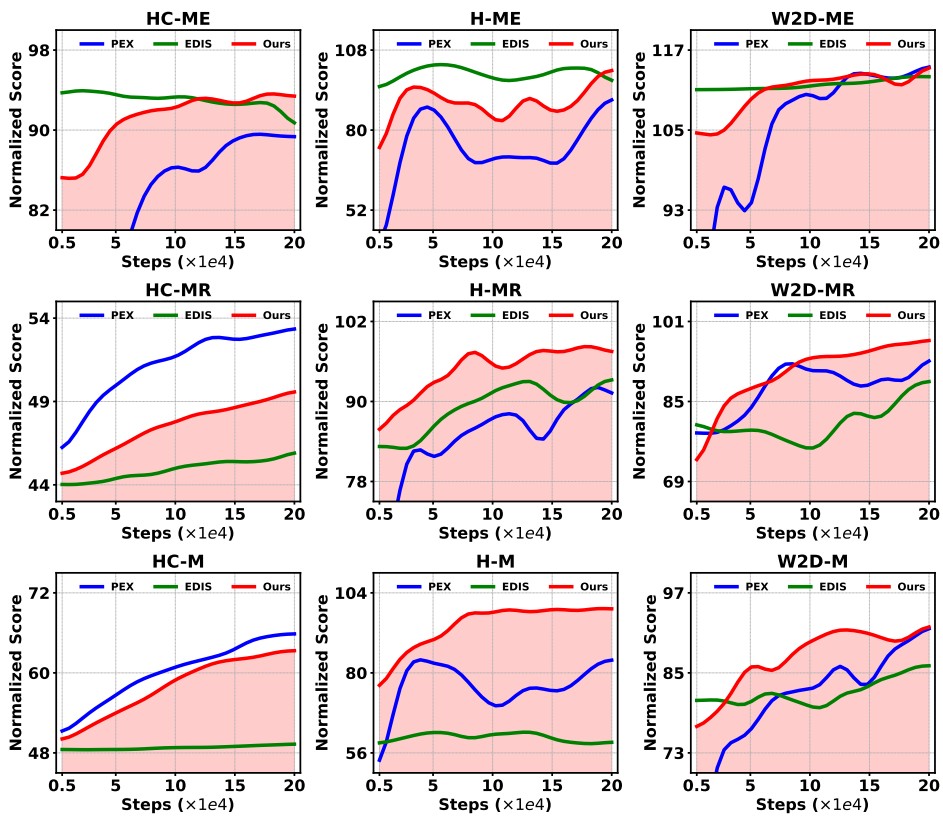

Figure 7: Online training curves on Locomotion benchmarks comparing PEX, EDIS, and Ours.

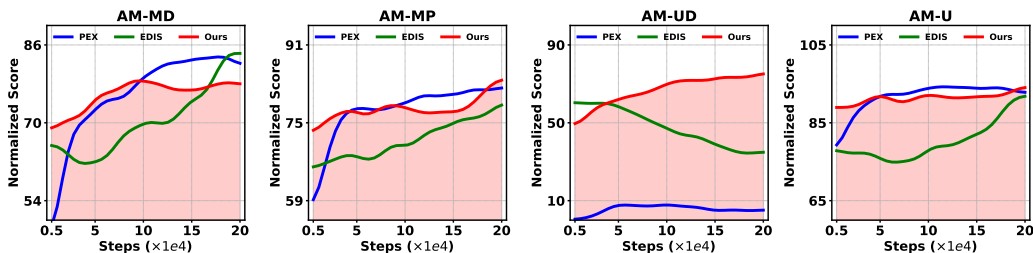

Figure 8: Online training curves on AntMaze benchmarks comparing PEX, EDIS, and Ours.

# D ADDITIONAL RESULTS ON EXCHANGE CAP

## D.1 EXCHANGE BEHAVIOR ACROSS RL BENCHMARKS

Fig. 9 shows the per-batch exchange statistics across environments. Consistent with Fig. 2, the realized number of swaps remains modest for most tasks (typically $< 70$).

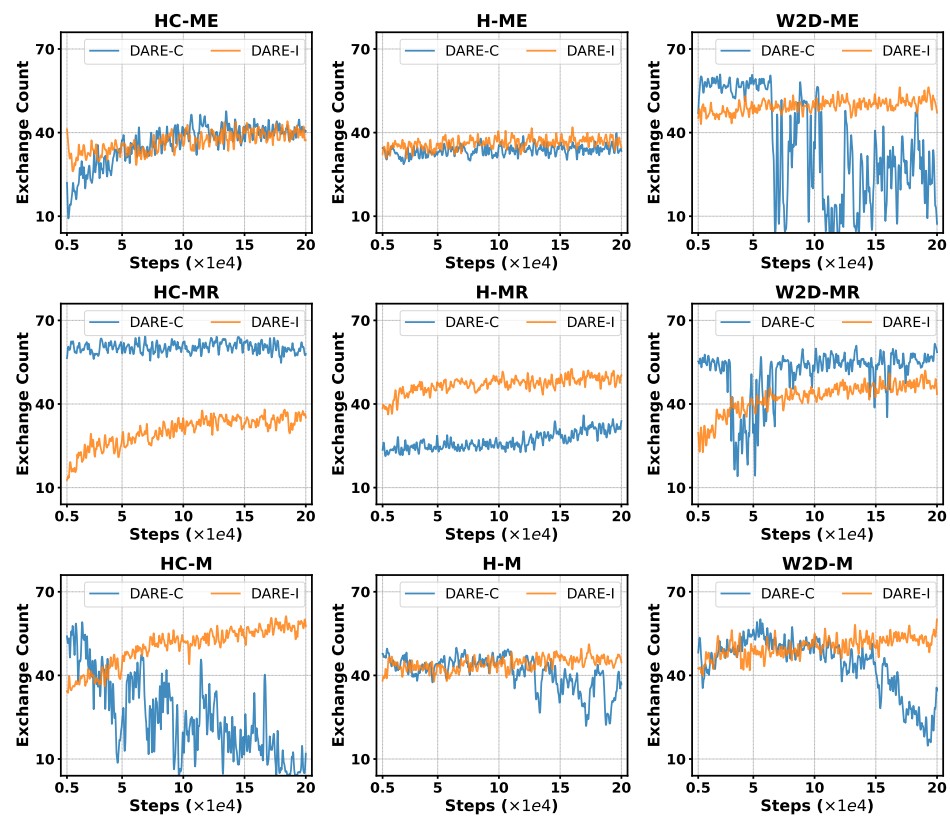

Figure 9: Per-batch exchange statistics across environments during online training (appendix).

## D.2 EFFECT OF THE MAXIMUM EXCHANGE CAP ON DARE-C

Tab. 9 extends Tab. 2 by adding $n=16$ for DARE-C. A moderate cap ($n=32$) gives the best overall performance, outperforming both a stricter cap ($n=16$) and the uncapped setting ($n=\infty$). Furthermore, Fig. 10 visualizes the training dynamics under different $n$.

|  | Dataset | $n = 16$ | $n = 32$ | $n = \infty$ |
|---|---|---|---|---|
| DARE-C | M | 255.9 | **265.9** | 258.5 |
|  | MR | 285.0 | **292.6** | 283.9 |
|  | ME | 320.1 | 320.2 | 320.3 |
|  | Total | 861.0 | **878.7** | 862.7 |

Table 9: Effect of the maximum exchange limit $n$ for DARE-C on MuJoCo Locomotion.

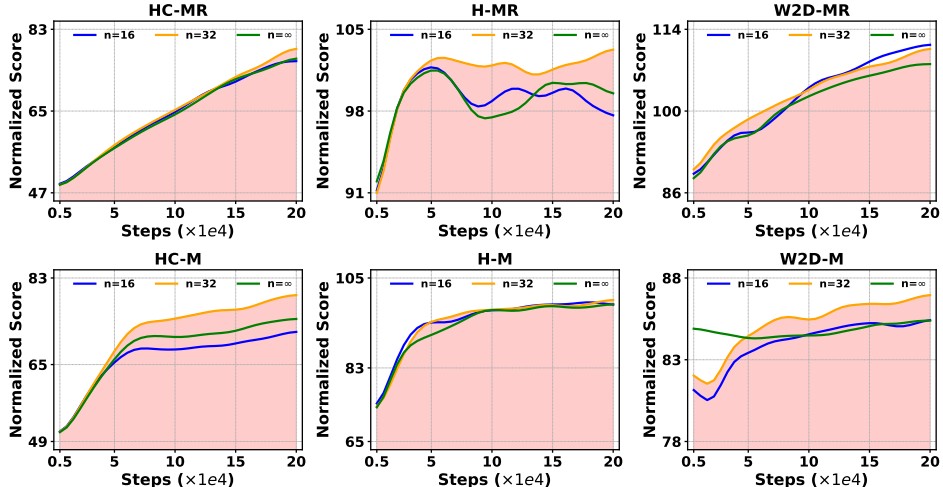

Figure 10: Training processes of DARE-C under different exchange caps $n$ on locomotion tasks.

### D.3 EFFECT OF THE MAXIMUM EXCHANGE CAP ON DARE-I

For DARE-I, we remove the cap consistently performed best across tasks. Capping at $n = 32$ does not improve asymptotic returns. We therefore use $n = \infty$ for DARE-I in all main experiments. Fig. 11 presents the training curves under different $n$, where the uncapped setting reaches higher performance.

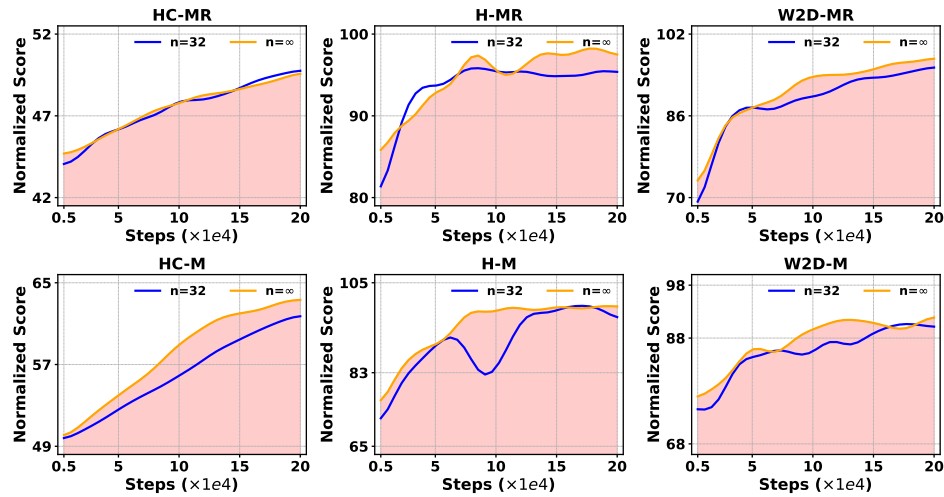

Figure 11: Training processes of DARE-I under different exchange caps $n$ (we select $n = \infty$ by default).

# E ADDITIONAL RESULTS FOR UPDATE-TO-DATA RATIOS

We extend Tab. 3 by adding the UTD = 1 results in Tab. 10. At both UTD = 1 and UTD = 10, DARE-I achieves the highest total after 0.1M online steps. In addition, the training curves for UTD = 10 in Fig. 12 show the same pattern.

Table 10: Performance of IQL-based methods under UTD = 1 and UTD = 10.

| Dataset | UTD = 1 | | | UTD = 10 | | |
|---|---|---|---|---|---|---|
| | PEX | EDIS-I | DARE-I | PEX | EDIS-I | DARE-I |
| HC-MR | **51.6±1.5** | 45.1±0.4 | 47.6±1.5 | **58.4±2.8** | 45.0±0.9 | 49.7±0.5 |
| H-MR | 86.6±9.6 | 92.9±11.0 | **94.9±3.9** | 88.5±22.8 | 94.1±9.7 | **100.2±2.4** |
| W2D-MR | 90.0±6.8 | 76.0±7.4 | **93.4±4.7** | 99.9±10.5 | 75.9±9.0 | **101.4±6.7** |
| HC-M | **60.9±1.9** | 48.8±0.2 | 59.0±1.1 | **76.1±1.6** | 48.8±0.2 | 66.3±1.4 |
| H-M | 68.3±17.3 | 61.1±5.3 | **97.6±7.9** | 84.6±21.4 | 60.8±5.3 | **101.2±4.3** |
| W2D-M | 84.3±15.2 | 79.3±8.3 | **89.7±7.4** | 92.5±18.2 | 81.4±5.5 | **99.1±1.6** |
| total | 442.2 | 403.2 | **482.2** | 495.8 | 406.0 | **517.9** |

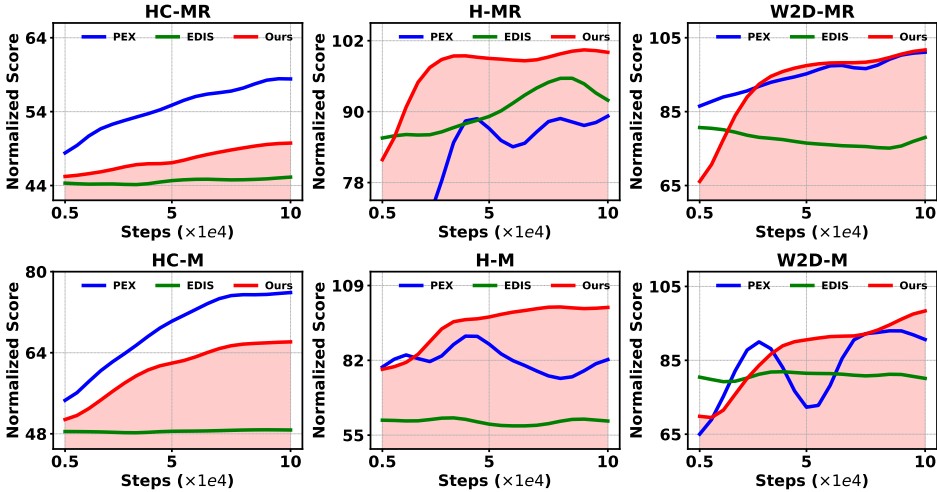

Figure 12: Online training processes comparison across Mujoco Locomotion tasks for UTD = 10 settings.

# F ADDITIONAL RESULTS FOR ABLATION STUDY

We present additional ablations to assess the contribution of the two key parts in DARE-C and DARE-I: energy guidance in the diffusion model and the sample exchange mechanism. As shown in Figs. 13 and 14, removing either part causes clear drops on MuJoCo Locomotion tasks.

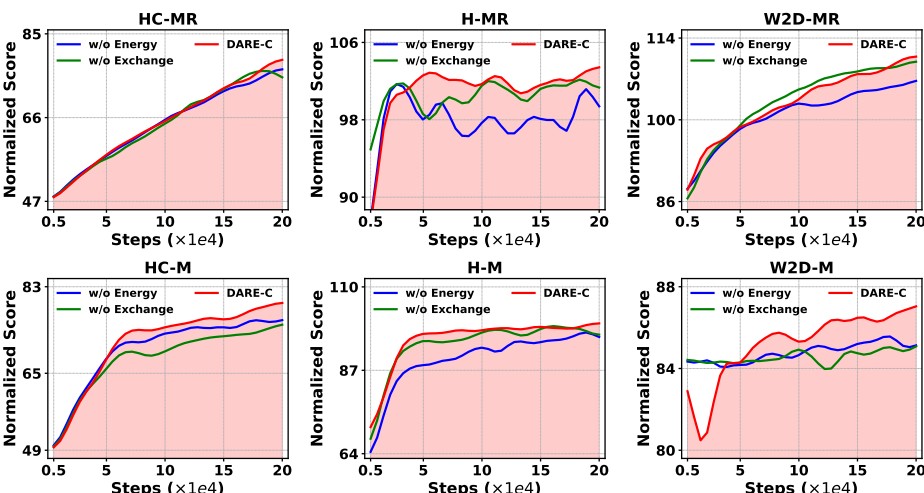

Figure 13: Ablation study for DARE-C.

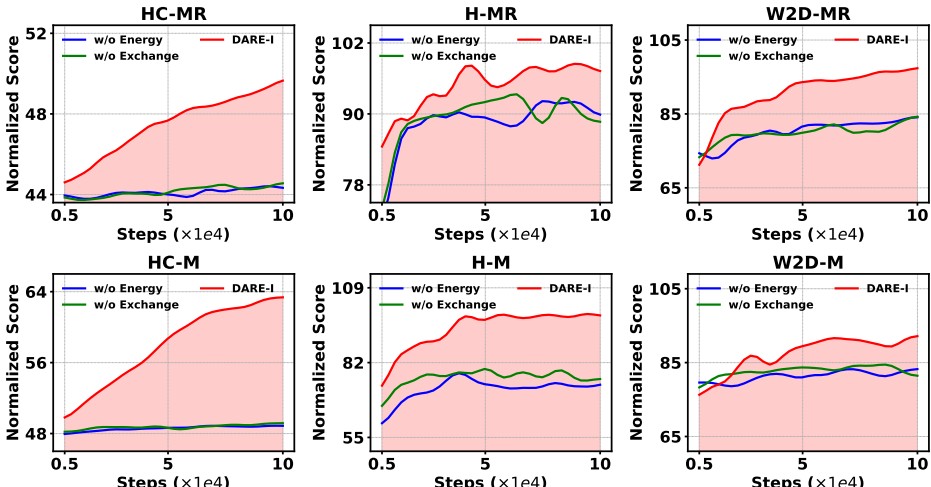

Figure 14: Ablation study for DARE-I.

## G    CHOICE OF ONLINE LOSS FOR IQL

We update IQL by applying SAC-style online losses on $b'_{\text{on}}$ while keeping standard IQL losses on $b'_{\text{off}}$ (full formulas in Sec. 6.1). This choice primarily targets the policy term.

For completeness, we also evaluate an alternative that keeps the same Q-target as in DARE-I (Eq. 15) while adjusts the value loss for online samples by using a higher expectile $\tau = 0.99$:

$$\mathcal{L}^{\text{IQL}}(V) = \mathbb{E}_{(s,a)\sim\mathcal{D}}\left[\mathbb{1}_{(s,a)\in b'_{\text{off}}} L_{\tau}^2(\hat{Q}(s,a) - V(s)) + \mathbb{1}_{(s,a)\in b'_{\text{on}}} L_{0.99}^2(\hat{Q}(s,a) - V(s))\right].$$
(35)

We refer to this variant as *DARE-IV*. Tab. 11 shows that *DARE-I* yields higher returns and lower variance. The same table also shows that our *sample-level constraint release* with batch exchange is effective when exchanges are enabled ($n = \infty$). Taken together, these results suggest that our strategy adapts well and integrates with different forms of constraint release.

Table 11: Performance of DARE-IV and DARE-I on MuJoCo Locomotion tasks.

| Dataset | DARE-IV | | DARE-I |
| --- | --- | --- | --- |
| | $n = \infty$ | $n = 0$ | $n = \infty$ |
| HC-ME | 91.9±2.8 | 92.0±2.1 | **93.5±1.0** |
| H-ME | 97.9±13.4 | 93.1±26.0 | **102.3±9.6** |
| W2D-ME | 113.2±1.2 | 111.2±0.8 | **114.7±1.1** |
| HC-MR | 48.1±1.3 | 45.5±1.3 | **49.4±1.0** |
| H-MR | 96.5±5.8 | 92.5±6.0 | **97.8±3.0** |
| W2D-MR | 96.3±6.5 | 86.2±5.2 | **97.0±3.1** |
| HC-M | 60.0±2.5 | 54.2±0.7 | **63.3±1.8** |
| H-M | 94.6±9.8 | 91.4±11.9 | **99.8±2.4** |
| W2D-M | **92.5±2.6** | 85.8±4.6 | 92.2±2.6 |
| total | 791.0 | 751.9 | **810.0** |

