# OpenReview forum: "From Static Constraints to Dynamic Adaptation: Sample-Level Constraint Release for Offline-to-Online RL"
_ICLR.cc/2026/Conference — ICLR 2026 Conference Withdrawn Submission_

### Official Review · Reviewer_ZJkf · 2025-10-21

**Soundness:** 2
**Presentation:** 2
**Contribution:** 2
**Rating:** 2
**Confidence:** 4

**Summary:**

This paper proposes Dynamic Alignment for RElease (DARE), an offline-to-online RL method that distinguishs offline-like and online-like samples and adaptively exchanges these samples between online and offline batches. Then relaxed or preserved constraints are imposed for online and offline batches, respectively. Theoretical results show that offline-online distributional discrepancy is reduced and the value estimation error is bounded. DARE also shows its empirical effectiveness on D4RL benchmark.

**Strengths:**

**1. Originality.** The observation that the offline and online distributions often overlap is interesting, and the idea that distinguishing the offline-like and online-like samples and exchanging them between offline and online batches seems novel.

**2. Significance.** DARE functions on data horizon and is compatible with different off2on methods such as Cal-QL and IQL. Experimental results show the effectiveness of DARE.

**Weaknesses:**

**1. On the clarity.** The writing and clarity of this paper is poor.  There are many typos, inconsistent writing, and concepts that lack clear explanations. To list a few:
- For text in formulas, please use \mathrm{} or \text{}. For instance, write $\\mathcal{D} _ \\mathrm{off}$ instead of $\\mathcal{D} _ {off}$ throughout the article.
- Please provide Figure 1(b) in PDF format to ensure image clarity.
- In Section 5.1, it is not explained how $\\mathcal{E} _ t(s _ t, a _ t)$ is derived from $Q(s,a)$ and how it is updated. Although Equation (34) in Appendix B.5 gives a contrastive learning objective to train the energy network, it is still unclear how this objective is related to $Q(s,a)$ and why it is reasonable.  Also, the lack of reference to Equation (34) in Section 5.1 makes it difficult for readers to understand this part.
- In Section 5.3, it should be made clear how $\\mu _ \\mathrm{off}, \\mu _ \\mathrm{on}, \\sigma _ \\mathrm{off}, \\sigma _ \\mathrm{on}$ are obtained.
- In Section 5.4, there are several symbols lack of explanations, such as $P$, $Q$ in Line 275, $Q^\\pi$ in Line 291. Also, only $d _ {H\\Delta H}$ and $\\mathcal{E} _ {\\alpha}$ are defined in Line 275 and Line 283, then why there are $d^{(0)} _ {H\\Delta H}$ and $\\mathcal{E}^{(0)} _ {\\alpha}$ with unexplained superscripts in Line 279 and Line 291. **Please make sure all the symbols are clearly explained before using them.** I strongly recommend the authors to double check the paper for these unclear concepts.

**2. On the theoretical analysis.** My major concerns are on the theorems proposed in theoretical analysis.
- Theorem 1 shows that the distributional discrepancy between offline and online samples is reduced with ​​more exchanges​​. However, this ​​contradicts​​ the motivation of the article. ​​If I understand correctly​​, the proposed exchange mechanism aims to distinguish offline-like from online-like samples, thus reducing the overlap as shown in Figure 1(c). In other words, the discrepancy between the offline and online distributions after ​​exchanges​​ should be larger, not smaller. Theorem 1 draws the opposite conclusion, which is quite confusing.
- Theorem 2 seems trivial and meaningless. The derivation of Theorem 2 relies on Proposition 3 (Bellman contraction) and the assumption that $|r _ Q|\\leq B _ T$. Then we can directly obtain $|\\hat{Q}-Q^{\\pi}|\\leq \\frac{B _ T}{1-\\gamma}$. Also, Theorem 2 shows that the estimation error bound is looser for a larger $M$. This seems confusing. Consider the following case, the original data distribution (A) changes to B after one exchage. Then the value estimation error bound under B would be larger than A. However, if we reverse the process, change B to A, then the error bound under A would be larger than B. This seems that the bound is not meaningful.

**3. Other concerns.**
- In Equation (7), it is imprecise to refer to the "KL divergence between actions," as the KL divergence is defined between probability distributions, not individual data points. Could you clarify how you compute Equation (7)?
- If we assume the offline and online scores follow Gaussian distribution, a more principled method is to construct a Gaussian Mixture Model (GMM) and use the EM algorithm to infer the probability that each data point is generated by different Gaussian distribution. Could you compare your method and EM algorithm?

**Questions:**

Please see the weaknesses for the concerns.

---

### Official Review · Reviewer_TQ48 · 2025-10-26

**Soundness:** 2
**Presentation:** 3
**Contribution:** 1
**Rating:** 2
**Confidence:** 4

**Summary:**

This paper addresses the core constraint-release dilemma in offline-to-online reinforcement learning, where conservative objectives from offline pretraining ensure stability but hinder adaptation, while uniform constraint removal causes instability, by proposing Dynamic Alignment for RElease, a distribution-aware framework that enforces sample-level, behavior-consistent constraints instead of the global constraints used in existing methods. Empirically, integrating DARE into two representative off2on methods (Cal-QL and IQL) and testing on the D4RL benchmark yields score improvements over baselines.

**Strengths:**

1. One strength lies in the rigorous, comprehensive experimental validation that robustly demonstrates DARE’s effectiveness.

**Weaknesses:**

1. The manuscript offers only incremental contributions. Personally speaking, earlier work does not introduce batch-level or global-level constraints; existing techniques, such as diffusion models or value-network-based importance sampling, apply constraints at the sample level. This submission does not present a genuinely new conceptual insight. It still enforces a constraint that forces the newly generated samples to align with the current policy, which is fundamentally the same mechanism already explored in prior studies.
2. The paper does not convincingly explain why a diffusion model is required. Since reference samples can be gathered directly from the current policy and distributional divergence can be estimated empirically, the diffusion model appears redundant and may introduce additional approximation errors and computational overhead. The authors should clarify under what conditions the diffusion model provides measurable advantages over direct sampling and quantify the bias–variance trade-offs it introduces.
3. Please also evaluate it on other suites, such as visual manipulation benchmarks or the Franka Kitchen tasks, in addition to the current experiments.

**Questions:**

See weaknesses.

---

### Official Review · Reviewer_9mee · 2025-11-01

**Soundness:** 2
**Presentation:** 1
**Contribution:** 2
**Rating:** 4
**Confidence:** 4

**Summary:**

This paper proposes a novel approach to mitigating the distribution shift challenge in offline-to-online reinforcement learning (off2on RL). The authors introduce a score-based mechanism that dynamically regulates the composition of offline and online batches and applies distinct training objectives to each based on behavioral alignment.

 Specifically, the proposed method, DARE (Dynamic Alignment for RElease), computes per-sample alignment scores using an energy-guided diffusion behavior model and distinguishes offline-like and online-like samples via Gaussian fitting. By selectively enforcing conservative constraints on offline-like samples while relaxing them for online-like ones, DARE aims to achieve a smoother transition from offline pretraining to online fine-tuning.

The paper further provides theoretical guarantees on bounded value estimation error and distributional alignment, and presents empirical results on D4RL benchmarks showing consistent and significant improvements over existing baselines such as Cal-QL and IQL. While the approach appears computationally demanding—since data must be manipulated for every batch—it presents a compelling and well-justified direction for addressing stability and adaptability in off2on RL.

**Strengths:**

**Strengths**

- The paper tackles the core challenge of distributional shift in offline-to-online RL with a principled and well-motivated approach.

- Proposes a sample-level constraint mechanism that adaptively rebalances offline and online data based on behavioral alignment, addressing the limitations of uniform constraint enforcement in prior work.

- The use of an energy-guided diffusion model and Gaussian fitting provides a data-driven and theoretically interpretable way to compute alignment scores and determine constraint boundaries.

- The method is algorithm-agnostic, demonstrated by seamless integration with both Cal-QL and IQL frameworks, and consistently improves performance across multiple D4RL benchmarks.

**Weaknesses:**

**Weaknesses**

1. Limited Experimental Domains
    - The experiments are restricted to relatively simple MuJoCo environments. The absence of evaluations on more complex domains raises concerns about the generalizability of the proposed approach. Including results on challenging benchmarks such as Kitchen, Adroit, or AntMaze-Large (as used in Cal-QL) would strengthen the paper's empirical credibility.
2. Concerns About Experimental Reliability
    - The reported baseline results in Table 1 appear inconsistent with those from the original papers. For instance, PEX achieves scores above 80 in the AM-MP setting according to its original publication, yet the authors report a score of 4.6. Similarly, the EDIS results deviate substantially from the reported numbers. It would be important to verify whether any implementation or configuration errors occurred in reproducing these baselines.
    - The proposed method's performance gains, while positive, do not appear statistically significant compared to the baselines. Given the added algorithmic complexity, the incremental improvements may not fully justify the overhead.
3. Ambiguity in Notation
    - The paper would benefit from improved mathematical clarity. For example, Equation (7) defines Alignment as the KL divergence between two samples, rather than between two distributions—it is unclear how this quantity is computed. Similarly, in Equation (10), the term $d$ is used to represent a score, but its exact definition is vague. Clarifying these aspects would greatly enhance readability and reproducibility.
4. Lack of Computational Cost Analysis
    - Since the proposed approach requires sampling actions from a diffusion model for each sample, it likely incurs nontrivial computational overhead. A comparison of computational costs against baselines would make the evaluation more complete and convincing.

**Miscellaneous Comments**

- The notation in Equations (14) and (15) is inconsistent; aligning these expressions would improve presentation quality.

**Questions:**

- Q1. Could you provide results on more challenging domains such as Kitchen or Adroit? These would help demonstrate the generalization capability of your method.

- Q2. How does the computational cost of your approach compare with that of the baselines? This is because the model seems expensive as it requires batchwise computation.

- Q3. Could you elaborate on how the score in (7) is precisely computed? A detailed formula or pseudocode would help readers understand the alignment mechanism more clearly.

- Q4. There are significant discrepancies between the baseline results in your paper and those reported in the original works. Could you clarify the reason for these differences?

- Q5. In your IQL-based online objective, you incorporate an SAC-style entropy term, whereas the offline objective lacks such entropy regularization. While you mention fixing the entropy coefficient, would the method remain stable under the common *target entropy* adjustment used in SAC? Please discuss the implications.

---

### Official Review · Reviewer_47S7 · 2025-11-03

**Soundness:** 3
**Presentation:** 2
**Contribution:** 2
**Rating:** 2
**Confidence:** 3

**Summary:**

This paper studies offline-to-online reinforcement learning. The paper argues that existing works fail to account for heterogeneity in the offline dataset and the online dataset. They propose DARE, an algorithm that puts online-like trajectories in the offline dataset to the online dataset, and offline-like trajectories in the online dataset to the offline dataset. In particular, they use KL divergence between an action and the behavior policy output to evaluate a trajectory's distance from the offline dataset, and label the trajectories accordingly. In the experiment part, they show superior performance against prior works.

**Strengths:**

1. The empirical performance is strong.

**Weaknesses:**

1. The paper is not well written enough for me to follow easily. See more in the questions below
2. I am not fully convinced that offline-like trajectory existing in online dataset should lead to performance drop, though the empirical performance seems strong. See more in the questions below.
3. The math formula are not rigorous in a lot of places. For example, I do not understand in formula 7 how to calculate the KL divergence of two vectors. In line 238 I assume score means the alignment score, however I thought it was the score for diffusion model whe I first read it. In formula 11 I do not understand which distributions are the metric evaluating. Overall I think there are too many parts that are not rigorous and is affecting my understanding.

**Questions:**

1. Could the authors further explain why having offline-like trajectories existing in the online dataset could lead to performance drop?
2. In Theorem 1 which distributions are we evaluating distance upon? Why do we want the offline and online distribution to be the same? Don't we want the online policy to concentrate on high reward regions?
3. In fomula 7, what does it mean by taking KL divergence between two actions?

---

### Note · Authors · 2025-11-12

I have read and agree with the venue's withdrawal policy on behalf of myself and my co-authors.